# *Pten* regulates collagen fibrillogenesis by fibroblasts through SPARC

Caitlin E. Jones[1]☯, Joe T. Sharick[1]☯, Sheila E. Colbert[1], Vasudha C. Shukla[1], Joshua M. Zent[1], Michael C. Ostrowski[2], Samir N. Ghadiali[1,3,4], Steven T. Sizemore[5,6], Jennifer L. Leight[1,6]*

**1** Department of Biomedical Engineering, College of Engineering, The Ohio State University, Columbus, Ohio, United States of America, **2** Department of Biochemistry and Molecular Biology, Hollings Cancer Center, Medical University of South Carolina, Charleston, South Carolina, United States of America, **3** Dorothy M. Davis Heart and Lung Research Institute, College of Medicine and Wexner Medical Center, The Ohio State University, Columbus, Ohio, United States of America, **4** Department of Internal Medicine, Division of Pulmonary, Critical Care and Sleep Medicine, College of Medicine and Wexner Medical Center, The Ohio State University, Columbus, Ohio, United States of America, **5** Department of Radiation Oncology, College of Medicine, The Ohio State University, Columbus, Ohio, United States of America, **6** The James Comprehensive Cancer Center, The Ohio State University, Columbus, Ohio, United States of America

☯ These authors contributed equally to this work.
* leight.1@osu.edu

**Data Availability Statement:** All relevant data are within the manuscript and its Supporting Information files.

**Funding:** Funding for this work was provided by the College of Engineering, The Ohio State

## Abstract

Collagen deposition contributes to both high mammographic density and breast cancer progression. Low stromal PTEN expression has been observed in as many as half of breast tumors and is associated with increases in collagen deposition, however the mechanism connecting PTEN loss to increased collagen deposition remains unclear. Here, we demonstrate that *Pten* knockout in fibroblasts using an *Fsp-Cre;Pten^{loxP/loxP}* mouse model increases collagen fiber number and fiber size within the mammary gland. *Pten* knockout additionally upregulated *Sparc* transcription in fibroblasts and promoted collagen shuttling out of the cell. Interestingly, *SPARC* mRNA expression was observed to be significantly elevated in the tumor stroma as compared to the normal breast in several patient cohorts. While SPARC knockdown via shRNA did not affect collagen shuttling, it notably decreased assembly of exogenous collagen. In addition, SPARC knockdown decreased fibronectin assembly and alignment of the extracellular matrix in an *in vitro* fibroblast-derived matrix model. Overall, these data indicate upregulation of SPARC is a mechanism by which PTEN regulates collagen deposition in the mammary gland stroma.

## Introduction

Collagen deposition within the mammary gland impacts both cancer development and progression. It is one of the primary tissue properties contributing to high mammographic density [1], which is one of the greatest risk factors for breast cancer development [2–4]. Furthermore, increasing collagen density in a murine model by mutating type I collagen to prevent cleavage by matrix metalloproteinases increased tumor development in the presence of oncogene expression [5]. In addition to collagen density, increasing collagen stiffness through

University and The James Comprehensive Cancer Center as startup funds to JLL, which also supported the salary of CEJ, JTS, and JMZ. JTS received salary support from the Ohio State University Presidential Postdoctoral Scholars Program. SNG, STS, and JLL received salary support from the Ohio State University. Images presented in this report were generated using the instruments and services at the Campus Microscopy and Imaging Facility, The Ohio State University. This facility is supported in part by NIH cancer center grant P30 CA016058 to The Ohio State University, National Cancer Institute, Bethesda, MD. The funders had no role in study design, data collection and analysis, decision to publish, or preparation of the manuscript.

**Competing interests:** The authors have declared that no competing interests exist.

crosslinking promotes tumor development and progression. Inhibition of the collagen-cross-linking lysyl oxidase (LOX) family in mice decreased fibrosis as well as tumor incidence and tumor size in a mouse model [6], and high lysyl oxidase-like 2 (LOXL2) expression correlates with low overall and metastasis-free survival in breast cancer patients [7]. While a number of single nucleotide polymorphisms have been identified which correlate with high mammographic density, including changes in fibroblast growth factor receptor 2 (FGFR2) and MAP3K1 [8], the molecular mechanisms connecting gene expression changes to collagen accumulation in cases of high mammographic density remain to be elucidated.

Stromal cells, particularly cancer-associated fibroblasts, are responsible for much of the collagen deposition in breast cancer. Low stromal PTEN expression is a widespread phenomenon in breast cancers, and has been observed in as many as half of patients [9]. We have previously shown that low stromal *Pten* expression correlates with high mammographic density in a small cohort of patients [10]. Knockout of the tumor suppressor *Pten* in fibroblasts in an *Fsp-Cre;Pten^{loxP/loxP}* mouse model increased both collagen deposition within the mammary gland and tumor development in mice expressing the *Neu* oncogene [9], which in combination with the patient data suggests that loss of stromal PTEN may contribute to the development of high mammographic density. However, the mechanism behind increased collagen deposition with PTEN loss is unclear. In this study, *Pten* knockout did not significantly affect transcriptional levels of fibrillar collagen, suggesting that other parts of the collagen deposition process are altered. Therefore, the question remains as to how *Pten* regulates this change in collagen deposition. We demonstrate here a new mechanism by which decreased PTEN expression upregulates secreted protein, acidic and rich in cysteine (SPARC) to regulate collagen fiber formation and organization.

SPARC is a matricellular protein which plays a role in collagen deposition and has previously been proposed as a collagen chaperone [11]. Knockout of *Sparc* in mouse models decreased collagen deposition within the dermis as well as collagen fiber size [12, 13]. Furthermore, SPARC knockdown greatly reduced bleomycin-induced lung fibrosis in a mouse model [14]. In solution, SPARC bound to a variety of different collagens, including collagen I, II, and III, and disrupted spontaneous collagen fibrillogenesis [15]. The mechanism of its effect on collagen secretion and assembly by cells remains to be elucidated.

Here, we demonstrate that *Pten* knockout in fibroblasts increases collagen secretion and SPARC expression. Furthermore, although SPARC knockdown in these fibroblasts did not affect collagen secretion, it decreased collagen fiber assembly. We further demonstrate that SPARC is important in fibronectin assembly and extracellular matrix alignment. Overall this implicates SPARC as a regulator of extracellular matrix assembly and organization, and suggests that SPARC contributes to the increased collagen deposition observed with PTEN loss in fibroblasts.

## Materials and methods

### Mouse studies

Wild-type (*Pten^{loxP/loxP}*) and *Pten^{-/-}* (*Fsp-Cre;Pten^{loxP/loxP}*) FVB/N mice were generated and maintained as previously described [9, 16] in compliance with federal and University Laboratory Animal Resources (ULAR) regulations and approved by the Ohio State University Institutional Animal Care and Use Committee under protocol 2007A0120-R1 (PI: MCO). Wild-type FVB/N were purchased from Jackson Laboratories (Bar Harbor, Maine, USA). All mice were 8-10wks of age and at least tenth-generation congenic (N10). Mice were housed in standard 12-hour light/dark cycles in individually ventilated microisolator cages with food and water

access ad libitum. Mice were euthanized by carbon dioxide inhalation followed by cervical dislocation. Glands were analyzed from 7 WT mice and 5 *Pten*$^{-/-}$ mice.

## Cell culture

Wild-type (WT; *Pten*$^{loxP/loxP}$) and *Pten*$^{-/-}$ (*Fsp-cre;Pten*$^{loxP/loxP}$) murine mammary fibroblasts cell lines were previously generated [9, 10, 17, 18]. Cells were used from passages 30–50 and maintained at 37°C and 5% $CO_2$ in high-glucose DMEM (Thermo Fisher Scientific) supplemented with 10% fetal bovine serum (FBS; VWR), 2 mM L-glutamine, 10 U/ml penicillin, and 10 μg/ml streptomycin. FBS was heat-inactivated in a water bath at 55°C for 30 minutes prior to use. Cells were tested for mycoplasma using the PromoKine PCR Mycoplasma Test Kit I/C (VWR) every three months.

## Antibodies and reagents

The following antibodies were used in these studies: collagen I (Abcam ab21286, used 1:1000 for WB, 1:250 for IF), SPARC (Cell Signaling 8725, clone D10F10, used 1:1000 for WB), GM130 (BD Biosciences 610822, clone 35/GM130, used 1:250 for IF), GAPDH (Thermo Fisher Scientific NB600502, clone 6C5, used 1:1000 for WB), tensin I (Novus Biologicals NBP1-84129, used 1:100 for IF), horseradish peroxidase-conjugated goat anti-rabbit or anti-mouse (Thermo Fisher Scientific 32460 and 32430, used 1:1000 for WB), and AlexaFluor-conjugated goat anti-rabbit or anti-mouse (Thermo Fisher Scientific A11034 and A21428, used 1:500 for IF). Alexafluor555-conjugated phalloidin (Thermo Fisher Scientific A34055, used 1:40 for IF) and Hoechst 33342 (Thermo Fisher Scientific H3570, used 1:1000 for IF) were used as counterstains.

## Second harmonic generation microscopy and collagen quantification

Second harmonic generation microscopy was performed as previously described [10]. In brief, 5 μm sections from the upper mammary gland of 8-week old mice were fixed and paraffin-embedded, then stained in Weigert's hematoxylin (Sigma-Aldrich) and picrosirius red (Abcam) according to the manufacturer's instructions. The sections were imaged in 1 μm slices through the depth of the tissue using an Olympus FV1000 MPE microscope with a 25× XLPlan water immersion lens (N.A. 1.05) and a Mai Tai DeepSee Ti:Sapphire laser (Spectra-Physics, Newport Corp.) at 950 nm wavelength. Maximum intensity z-projections of the images were analyzed using CT-Fire software (v1.3, https://loci.wisc.edu/software/ctfire) [19] to determine collagen fiber length and width.

## shRNA knockdown

Stable SPARC knockdown lines were produced using MISSION lentiviral transduction particles (Sigma-Aldrich) with a pLKO.1-puro vector, using a multiplicity of infection of 10. Five different shRNA sequences were tested and the two with the highest knockdown efficiency were chosen to carry forward. The shRNA sequences used are as follows: 5'-CCGGCCTAGACA ACGACAAGTACATCTCGAGATGTACTTGTCGTTGTCTAGGTTTTTG-3' (TRCN0000080349, designated shSPARC#1) and 5'-CCGGGAATACATTAACGGTGCTAAACTCGAGTTTAGCACC GTTAATGTATTCTTTTTTG-3' (TRCN0000418600, designated shSPARC#2). A non-targeting scrambled sequence was used as a control (Sigma-Aldrich SHC016V). Cells were transfected at 30–40% confluence and pretreated with 8 μg/mL polybrene for 45 minutes prior to viral addition. Transfection was performed for 24 hours, then the cells were allowed to recover for 24 hours prior to placing under selection with 5 μg/mL puromycin for 3 days.

## qRT-PCR

Cells were plated for 24 hours and then supplemented with 50 μg/mL L-ascorbic acid (Sigma-Aldrich) for a further 24 hours prior to lysis. RNA was isolated using Tri-Reagent (Sigma-Aldrich) using the manufacturer's instructions, then reverse transcribed using the iScript DNA synthesis kit (Bio-Rad). qRT-PCR was run using primers at 900 nM (Table 1), and results were normalized to the endogenous control, 18S (Thermo Fisher Scientific, primer pair used at 100 nM). Primer sequences were obtained from the Harvard Primer Bank (https://pga.mgh.harvard.edu/primerbank/) and verified using Primer BLAST (https://www.ncbi.nlm.nih.gov/tools/primer-blast/), or designed using Primer BLAST.

## Western blotting

Cells were plated in 10% heat-inactivated FBS DMEM. After 4 hours, the cells were washed with PBS and the medium replaced with DMEM and 50 μg/mL L-ascorbic acid (Sigma-Aldrich). After 20 hours, the medium was collected and the cells were lysed in RIPA buffer containing 25mM Tris-HCl pH 7.6, 150 mM NaCl, 1% NP-40, 1% sodium deoxycholate, and 0.1% SDS (Thermo Fisher Scientific), with 1x HALT protease/phosphatase inhibitor cocktail (Thermo Fisher Scientific). The samples were frozen at -70˚C until further use. Lysates were centrifuged at 12000RPM for 15 minutes to pellet debris, and the concentration of protein in the supernatant was quantified using the micro BCA assay kit (Thermo Fisher Scientific). The cell culture medium was concentrated using 10 kDa molecular weight cutoff centrifugal filters (Millipore-Sigma). 25 μg of protein was used for each sample of the cell culture lysate, while for the cell culture medium equal volumes of medium was used for each sample. Samples were treated with 100 mM dithiothreitol (Sigma-Aldrich) for 15 minutes, then the Bolt LDS sample buffer (Thermo Fisher Scientific) was added and the samples were heated at 70˚C for 10 min.

**Table 1. Primer sequences.**

| Gene | F | R |
|---|---|---|
| Sparc | TGGGAGAATTTGAGGACGGTG | GAGTCGAAGGTCTTGTTGTCAT |
| Col1a1 | GCTCCTCTTAGGGGCCACT | CCACGTCTCACCATTGGGG |
| Col1a2 | GGTGAGCCTGGTCAAACGG | ACTGTGTCCTTTCACGCCTTT |
| Col3a1 | CTGTAACATGGAAACTGGGGAAA | CCATAGCTGAACTGAAAACCACC |
| Col4a1 | TCCGGGAGAGATTGGTTTCC | CTGGCCTATAAGCCCTGGT |
| Lox | ACTTCCAGTACGGTCTCCCG | GCAGCGCATCTCAGGTTGT |
| Loxl1 | CGCCCTTCGTAAACCAGTATG | CACCACGGTAGTACACGTAGC |
| Loxl2 | TCGGAGCTTTTCTTCTGGGC | GGAGAGGCAGGAGCCAAAAT |
| Loxl3 | AGGGCCTGGTTGAAGTCAG | CCAGCATACCGCAGACTACG |
| Loxl4 | GTGCCAAGTATGGTCAAGGAG | CCGTTAGAGCCACACTGATCT |
| P4HA1 | AGACCGGCTAACGAGTACAG | CCAACTCACTCCACTCAGTGT |
| P4HA2 | CACCTCCATTGGGCACATGA | GCTCTTAATCTTGGCGAGCTT |
| p4HA3 | AGACGCCCTGGATTACTTGG | GTCTGGGCTGTAGACAAGGAA |
| Plod 1 | AGGATGACGCCAAGCTAGAG | TTGAAGAACTGAGCTGAACGC |
| Plod 2 | GCTGGCAGACAAGTACCCTG | GGGGCATAGCCAATAAAGCC |
| Plod 3 | AAGAAGTTTGTTCAGAGTGGCA | TGAATCCACCAGAGTTGAGGA |
| SOX5 | GTTGGAGCCAGCCGATCATA | ACCTCTCCATCTGTCTCCCC |
| KLF4 | TACCCCTACACTGAGTCCCG | GGAAAGGAGGGTAGTTGGGC |
| Brg1 | ACAAAGACAGCAGCAGTGGA | AGTCCACAGGCTTTCGGATG |
| Snai1 | GAGCCTCCTACCCCTCAGTA | AGCCAGTGGGTTGGCTTTAG |

Samples were separated using Bolt 4–12% Bis-Tris Plus gels and Bolt MOPS-SDS running buffer, and transferred onto a PVDF membrane using Bolt Transfer Buffer (Thermo Fisher Scientific). Precision Plus Protein Dual Color Standard (Bio-Rad) was used as a molecular weight ladder. Signal was detected using an HRP-conjugated secondary antibody and chemiluminescent substrate (SuperSignal West Femto for collagen, SuperSignal West Pico PLUS for SPARC and GAPDH, Thermo Fisher Scientific). ImageJ software (National Institutes of Health, v1.43u) was used to perform densitometry.

## Immunostaining and imaging

For immunostaining of collagen and GM130, cells were fixed in 4% paraformaldehyde (Electron Microscopy Sciences) for 20 minutes and permeabilized in 0.5% Triton X-100 in PBS for 5 minutes. For immunostaining of tensin, cells were simultaneously fixed and permeabilized using 3% paraformaldehyde with 0.5% Triton-X 100 in PBS for 2 minutes, followed by 3% PFA in PBS only for 30 minutes [20]. All cells were then blocked using 10% normal goat serum in PBS (NGS, Invitrogen) for 30 minutes. Primary antibodies were added at the indicated concentration in 10% NGS in PBS for 1 hour at room temperature, then the samples were washed 3x10 minutes in PBS. Secondary antibodies were added in 10% NGS in PBS for 1 hour at room temperature prior to washing 3x10 minutes in PBS and mounting the samples using ProLong Gold Antifade solution (Life Technologies). Samples were imaged at room temperature using an Olympus FV1000 confocal microscope equipped with a 20× (N.A. 0.85), 40× (N.A. 1.3), or a 60× (N.A. 1.4), oil-immersion objective lens.

## Image analysis

A custom CellProfiler routine was written to automatically detect individual cell cytoplasms and extract the integrated density of collagen immunofluorescence for each [21]. Briefly, nuclei were detected using Hoechst stain signal, and cell borders were detected using the phalloidin actin stain. Both signals were thresholded using the minimum cross-entropy method. Nuclei regions were subtracted from cell borders to create individual cytoplasm masks.

Individual fibrillar adhesions were detected using CT-Fire. Briefly, tensin images were thresholded to create a binary mask, and minimum "fiber" length was set to 3.5μm. This cutoff was chosen by generating a histogram of lengths of all detected fibers, which revealed a second small peak above this length (S2 Fig). This length is also similar to previous measurements of average fibrillar adhesion length in fibroblasts [22]. The ratio of number of fibrillar adhesions in each field of view to the total integrated density of tensin signal in the field of view was calculated. Each biological replicate was normalized to the average output of all images taken on that day in order to control for day-to-day variation in signal intensity.

## Collagen and fibronectin fibril assembly

For matrix assembly studies, fluorescently labeled collagen (10 μg/mL) or fibronectin (6 μg/mL) was added to cells plated 24 hr after plating. Cells were cultured in growth medium in the absence of ascorbic acid. After 24 hr culture with the fluorescently labeled protein, cells were fixed and stained as described above.

To fluorescently label collagen, 1 mg rat tail type I collagen (1 mg/mL in PBS; Corning) was incubated for 2 hours on ice, rocking, with 12.5 μL of 1M sodium bicarbonate buffer, pH 9, and 12.5 μL of 1 mg/mL NHS-Fluorescein (Fisher Scientific) in DMSO. The collagen was then placed in 8 kDa MWCO dialysis tubing and left in 0.1% acetic acid in PBS for 4 hours at 4˚C, stirring, to remove unconjugated NHS-Fluorescein. The labeled collagen was stored at 4˚C.

Fibronectin was labeled with NHS-Fluorescein according to previously a published proto-col [23], with some minor modifications. Briefly, 10 mg of bovine fibronectin (Millipore-Sigma) was suspended at 1 mg/mL in PBS. Fibronectin was dialyzed in PBS overnight using 8 kDa dialysis tubing, then incubated with 125 μL of 1M sodium bicarbonate buffer, pH 9, and 125 μL of 1 mg/mL NHS-Fluorescein (Fisher Scientific) in DMSO for 2 hours at room temper-ature, rocking. Labeled fibronectin was separated using PD-10 desalting columns (GE Life Sci-ences) and stored at -70˚C.

## Fibroblast-derived matrix production

Fibroblast-derived matrices were produced as previously described [24]. Briefly, cells were plated on gelatin-coated coverslips for 24 hours, then on day 2 and day 4 of the experiment the medium was replaced and supplemented with 50 μg/mL ascorbic acid and 6 μg/mL fluores-cently labeled fibronectin. Cells were fixed on day 6 by removing half of the culture medium and replacing it with a solution containing 4% paraformaldehyde and 5% sucrose. Nuclei were stained with Hoechst and the matrices were imaged in 8 μm stacks through the matrix in 1 μm intervals. The images were Z-projected and fiber orientation was quantified using the ImageJ plugin OrientationJ [25] with previously published settings [24]. The mode fiber orientation was set to 0˚ and matrix alignment was quantified by determining the fraction of fibers falling within 10˚ or 20˚ of the mode matrix orientation.

## Traction force microscopy

Traction force microscopy was used to measure the contractility of the fibroblasts at individual cell level as described previously [26]. Cells were maintained under normal culture conditions. TFM substrates were prepared with polyacrylamide (PA) on 22 mm glass coverslips using an 8% acrylamide and 0.04% bis-acrylamide solution (BioRad) polymerized with 10% ammonium per-sulphate and tetramethylethylenediamine (TEMED). The gels were embedded with 0.5 μm-diameter red fluorescent carboxylate modified beads (Invitrogen) and coated with 50 μg/mL bovine collagen type I (Advanced BioMatrix Inc.). Cells were seeded (~1000/gel) in their respec-tive treatment media and allowed to adhere overnight. After 24 hours, isolated single cells were imaged using phase contrast microscopy to identify cell boundaries, while fluorescent images of bead positions were obtained before and after cell detachment by trypsinization. Images were then analyzed to compute the bead displacement along the cell boundary using correlation-based particle image velocimetry in MATLAB (MathWorks). The traction stress field on the gel surface was calculated using the 3D finite element software COMSOL Multiphysics where the measured bead displacements under the cell were applied as a boundary condition, and the resulting stress field was analyzed for average traction and net contractile moment [26].

## Patient datasets

Results from this screen were compared to several publicly available stromal gene expression datasets including those generated by Finak *et al* [27] Gene Expression Omnibus (GEO): GSE9014, Karnoub *et al* [28] GEO:GSE8977, and Ma *et al* [29] GEO:GSE14548 for compari-son between normal and breast tumor-associated stroma expression. mRNA expression levels were compared by t-test.

## Statistical analysis

Statistical testing was done using GraphPad Prism v7 and v8. Comparisons between two groups were performed using a two-tailed unpaired student's t-test. Comparisons between

more than two samples were performed using a one-way ANOVA followed by Tukey's post-test. Comparison of collagen fiber length and fiber width between conditions was done by pooling fibers from all samples into a histogram and using a two-sample Kolmogorov-Smirnov test to examine differences in sample distribution. Differences were considered significant at $p < 0.05$.

## Results

### *Pten* knockout increases collagen fiber formation and fiber size in the mammary gland

While PTEN is well known for its role as a tumor suppressor, it has more recently been identified as a regulator of extracellular matrix deposition and organization. Increases in collagen deposition within the mammary gland have previously been reported with *Pten* knockout using an *Fsp-Cre;Pten^{loxP/loxP}* mouse model, particularly increases in collagen I [9]. To characterize the changes in collagen within the mammary gland, second harmonic generation microscopy was used to image collagen I surrounding ducts within the mammary gland of mice with WT or *Pten^{-/-}* fibroblasts (Fig 1A). There was a significant increase in the number of

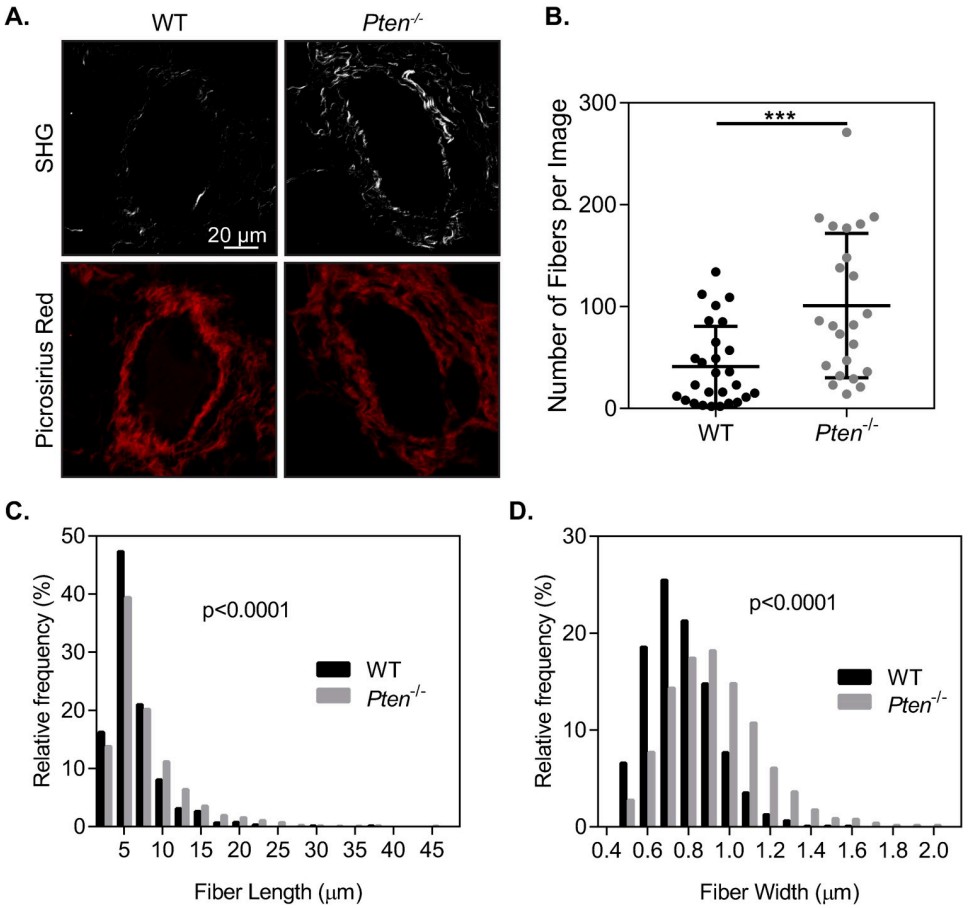

**Fig 1. Pten knockout increases collagen fiber formation.** (A) Second harmonic generation (SHG) and picrosirius red staining surrounding ducts in murine mammary tissue sections. SHG images are shown above their corresponding picrosirius red images. (B) Quantification of the number of collagen fibers in SHG images, shown as mean+SD with individual values plotted, n = 32 ducts imaged across 7 WT mice, 23 ducts imaged across 5 *Pten^{-/-}* mice. (C) Histogram of collagen fiber length in SHG images. (D) Histogram of collagen fiber width in SHG images. ***$p < 0.001$.

collagen fibers surrounding mammary ducts in the *Pten*$^{-/-}$ glands (Fig 1B). Collagen fiber width and length were quantified using CT-Fire software [19] and plotted as a histogram (Fig 1C and 1D). There was a small but statistically significant shift in the distribution toward longer fiber lengths, as well as a significant (p<0.0001) shift in the distribution of fiber width, indicating an increase in collagen fiber size in the *Pten*$^{-/-}$ mammary glands. Overall, this suggests that there may be changes in collagen fibrillogenesis with *Pten* knockout, although it remains unclear whether this is due to changes in collagen production or assembly.

### PTEN loss alters collagen processing

Deposition of a collagen matrix is regulated at multiple levels, including collagen synthesis, secretion, and fiber assembly. However, it is unknown which of these levels of regulation are impacted by Pten, leading to the observed increased collagen deposition by *Pten*$^{-/-}$ fibroblasts in the mammary gland. Collagen is initially synthesized as a propeptide which requires hydroxylation for protein folding and transportation from the endoplasmic reticulum (ER) to the Golgi apparatus [30]. After secretion into the extracellular space, proteases such as BMP-1 [31] and members of the ADAMTS family [32] cleave the C- and N-terminal peptides and trigger self-assembly of collagen fibers [33, 34]. More recently, cell contractility and $\alpha_2\beta_1$ integrin engagement [35], as well as assembly of a pre-existing fibronectin matrix [36] have been shown to contribute to collagen fiber assembly.

To determine which part of this collagen processing pathway is altered with *Pten* knockout, transcriptional levels of collagen genes were first examined in *Pten*$^{-/-}$ murine mammary fibroblasts as compared to WT cells (Fig 2A). While the fibrillar collagen genes *Col1a1*, *Col1a2*, or *Col3a1* were not significantly different between the two cell types, *Col4a1* mRNA levels were significantly increased with *Pten* knockout. Given the changes in collagen fiber number and size observed within the mammary tissue, the lack of large changes in fibrillar collagen mRNA levels suggests that collagen deposition is altered post-transcriptionally. Collagen protein expression was analyzed by western blot in WT and *Pten*$^{-/-}$ fibroblasts supplemented with 50 μg/mL ascorbic acid for 20 hr. Ascorbic acid is an essential cofactor for prolyl-4-hydroxylases to hydroxylate collagen as part of the folding process prior to leaving the ER; in the absence of ascorbic acid, collagen is retained in the ER [30, 37]. Collagen protein expression was measured in the cell lysate and concentrated cell culture medium to observe both intracellular and extracellular collagen levels (Fig 2B, full blot in S1 Fig). Samples were treated with dithiothreitol to ensure solubilization of the collagen. While expression was somewhat variable, protein expression of collagen was not statistically different between the WT and *Pten*$^{-/-}$ fibroblasts in the cell lysate or culture medium at this time point (Fig 2C and 2D). Collagen immunofluorescence of cells cultured with ascorbic acid for 20 hr showed no visible collagen deposited extracellularly at this time point (S3 Fig).

Collagen expression can also be regulated by secretion and the shuttling of collagen from the ER to the Golgi apparatus and into the extracellular space. To observe changes in collagen transport within the cell, cells were plated for 24 hours and supplemented with ascorbic acid for 1 hour prior to fixation. The cells were then stained for collagen and GM130, a Golgi marker (Fig 3A). In the absence of ascorbic acid, collagen appeared to be retained in the endoplasmic reticulum and showed no noticeable colocalization with the Golgi in either cell type, as expected since ascorbic acid is necessary for collagen shuttling from the ER. In the presence of ascorbic acid, the *Pten*$^{-/-}$ cells had significantly increased collagen colocalization with the Golgi as compared to either the WT cells or the *Pten*$^{-/-}$ cells without ascorbic acid supplementation (Fig 3B). In contrast, ascorbic acid addition did not noticeably increase collagen colocalization with GM130 in WT cells, suggesting that collagen remained in the ER. These data indicate that collagen shuttling from the ER to the Golgi is increased with *Pten* knockout [10].

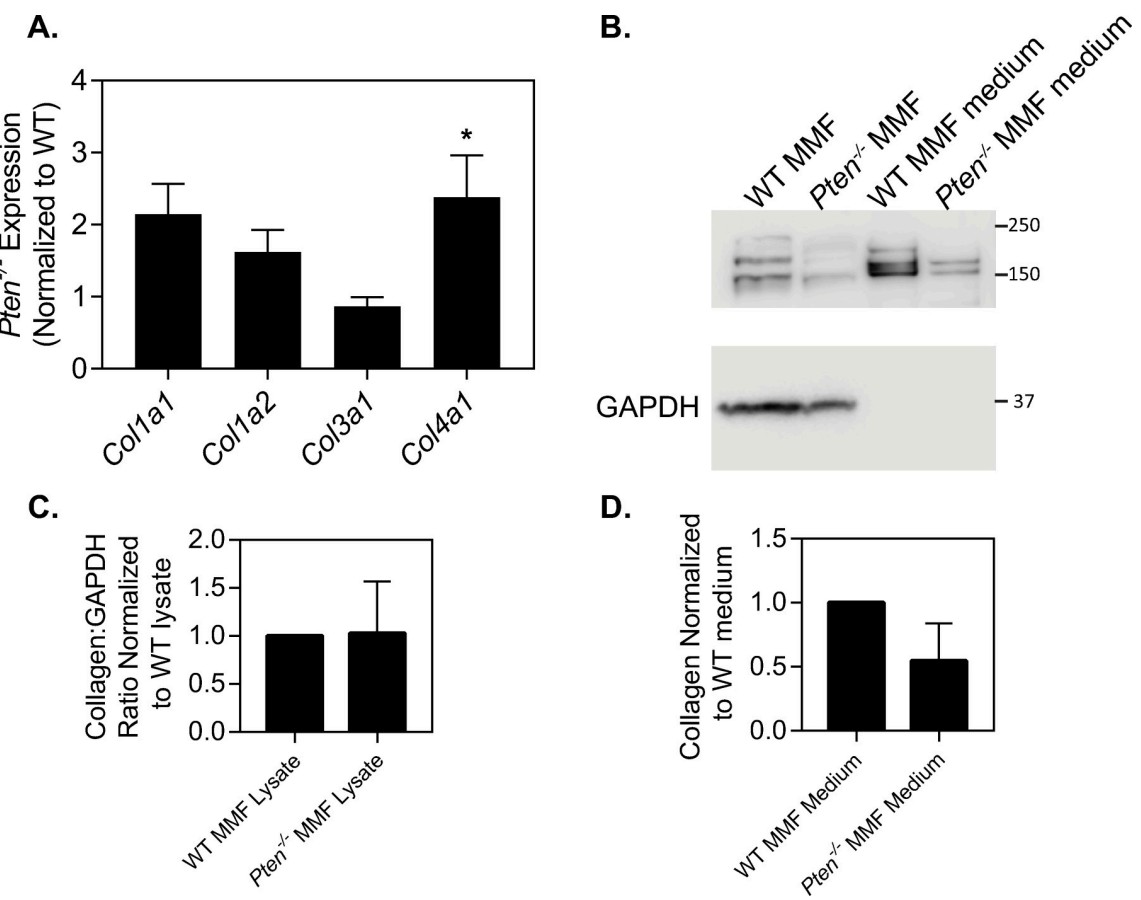

**Fig 2. *Pten* knockout does not change collagen expression.** (A) Normalized collagen mRNA expression, n = 5+SD. (B) Western blot of collagen I, with GAPDH used as a loading control for cell lysates. Individual bands likely represent procollagen, processing intermediates, and alpha subunit chains. (C) Quantification of collagen in cell lysates, normalized to WT. n = 3+SD. (D) Quantification of collagen in culture medium, normalized to WT. n = 3+SD. *p<0.05.

We next hypothesized that Pten may also be altering collagen fibrillogenesis after collagen secretion by the cell. To test this, fluorescently labeled type I collagen and fluorescently labeled fibronectin were added exogenously to cells in the absence of ascorbic acid, to isolate assembly by the cells from the ability of these cells to produce and secrete these proteins (Fig 3C). Fibrillogenesis was quantified using ImageJ to examine the area within each image covered by collagen and fibronectin, relative to the number of nuclei in the image (Fig 3D and 3E).

## PTEN loss alters collagen processing genes

In order to further elucidate the changes to collagen processing with *Pten* knockout, we used RT-PCR to examine mRNA expression of different genes involved in collagen processing. The LOX family is involved in collagen crosslinking outside of the cell [38], while prolyl-4-hydroxylases (*P4ha1*, *-2*, and *-3*) and lysyl hydroxylases (*Plod1*, *-2*, and *-3*) hydroxylate procollagen to induce correct folding and are necessary for procollagen to leave the ER [37]. SPARC is important for collagen deposition and its attenuation via siRNA significantly decreases fibrosis [14] but the exact mechanism of its effect on collagen is not yet understood. *P4ha3* and *Sparc* were significantly upregulated in *Pten*⁻ᐟ⁻ fibroblasts compared to WT fibroblasts (Fig 4A). P4HA3 hydroxylates collagen in a similar manner as the other two prolyl-4-hydroxylase isoforms, but

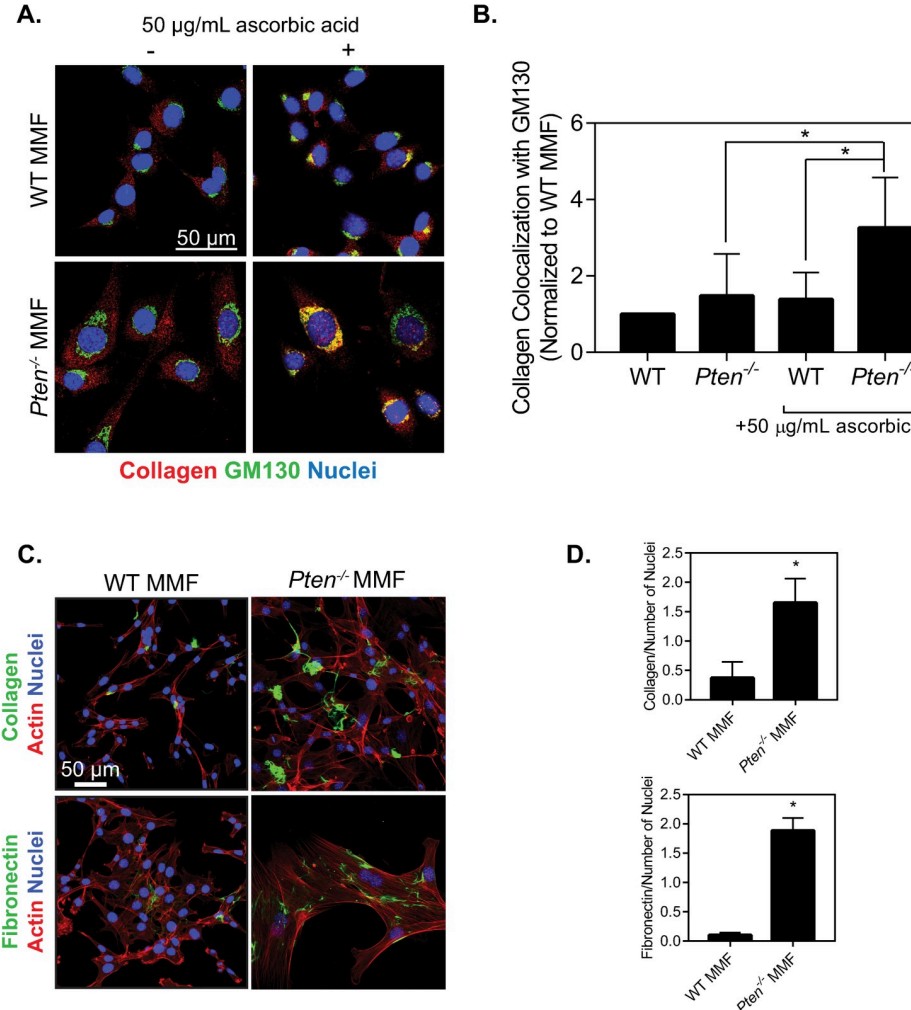

**Fig 3. *Pten* knockout alters collagen shuttling and fibrillogenesis.** (A) Immunofluorescence images of WT and *Pten*−/− MMF plated for 24 hours and cultured with or without 50 μg/mL ascorbic acid for 1 hour, then fixed and stained for collagen, GM130, and nuclei. (B) Quantification of collagen colocalization with GM130, determined by Manders' correlation coefficient and normalized to the WT condition. n = 5+SD. *p<0.05. (C) Immunofluorescence images of cells incubated with fluorescently labeled collagen and fibronectin for 24 hours, then fixed and stained for actin and nuclei. (D) Quantification of the area of each image covered by collagen fibers normalized to number of nuclei in each image. (E) Quantification of the area of each image covered by fibronectin fibers normalized to number of nuclei in each image. n = 3+SD. *p<0.05 relative to WT.

is typically expressed at much lower levels in most adult tissues [39]. As the role of prolyl-4-hydroxylases in collagen synthesis is well-characterized [40–42], we chose to examine the role of SPARC in *Pten* knockout-induced collagen deposition. *Pten*−/− fibroblasts showed a notable (~12-fold) increase in SPARC expression and in SPARC secretion (~4-fold) into the extracellular space versus WT cells (Fig 4B, quantified in 4C, full blot in S1 Fig). As SPARC appeared to be increased transcriptionally, the expression of several known transcription factors affecting SPARC was examined by RT-PCR (Fig 4D). The expression of *Sox5*, a known transcriptional repressor of *Sparc* [43], was decreased ~150-fold in *Pten*−/− cells. Of several known factors which directly transcribe *Sparc* [44–46], the expression of *Klf4* was significantly increased.

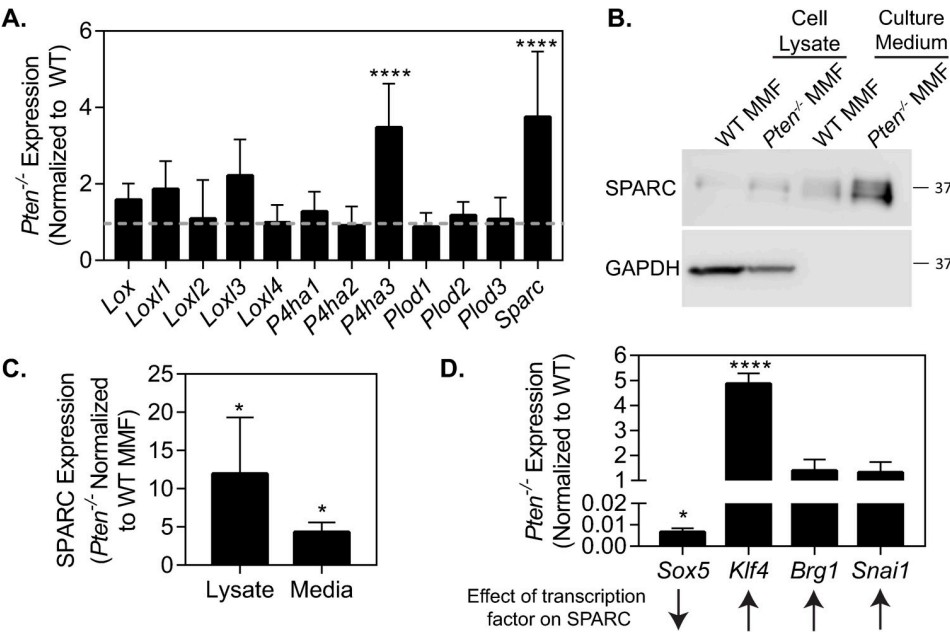

**Fig 4.** ***Pten*** **loss upregulates SPARC expression.** (A) mRNA expression of collagen-processing proteins. n = 5+SD. (B) Western blotting for SPARC expression, with GAPDH as a loading control. (C) Quantification of SPARC expression in cell lysate and culture medium, normalized to WT. n = 3+SD. (D) mRNA expression of transcription factors known to affect SPARC. n = 3+SD. *p<0.05, ****p<0.0001

## SPARC knockdown does not affect collagen secretion by *Pten⁻ᐟ⁻* fibroblasts

To examine the effects of SPARC on collagen trafficking, SPARC was knocked down in *Pten⁻ᐟ⁻* fibroblasts using shRNA. Analysis of the knockdown via Western blot showed ~80% knock-down in in both the cell lysate and in the culture medium in two lines (Fig 5A, full blot shown in S1 Fig, quantified in S4 Fig). To determine how SPARC knockdown affected collagen secre-tion, collagen protein expression in the cell lysate and culture medium were measured using a Western blot. (Fig 5B, full blot in S1 Fig). Although we hypothesized that SPARC knockdown would decrease collagen secretion, no collagen retention was visible within the cell and colla-gen levels in the culture medium were similar across cell lines (Fig 5C). Similarly, we examined collagen colocalization with the Golgi apparatus in the presence of ascorbic acid, and saw no significant change in colocalization, although all cell lines showed a significant response with ascorbic acid addition (Fig 5D and 5E). Overall, this indicates that SPARC does not play a notable role as a collagen chaperone in this model. We therefore hypothesized that the change in collagen shuttling to the Golgi with *Pten* knockout may be due to the observed increase in *P4ha3* levels (Fig 4A).

## *Pten* knockout-induced upregulation of SPARC promotes ECM assembly

As SPARC expression did not appear to affect collagen trafficking out of the cell, we therefore hypothesized that SPARC may be affecting collagen fibrillogenesis after collagen secretion by the cell as SPARC is a known regulator of collagen fibrillogenesis [12, 13, 15, 47]. To test this hypothesis, fluorescently labeled type I collagen was added exogenously to cells in the absence of ascorbic acid, to isolate collagen assembly by the cells from the ability of these cells to pro-duce and secrete collagen (Fig 6A). Fibrillogenesis was quantified using ImageJ to examine the area within each image covered by collagen, relative to the number of nuclei to control for

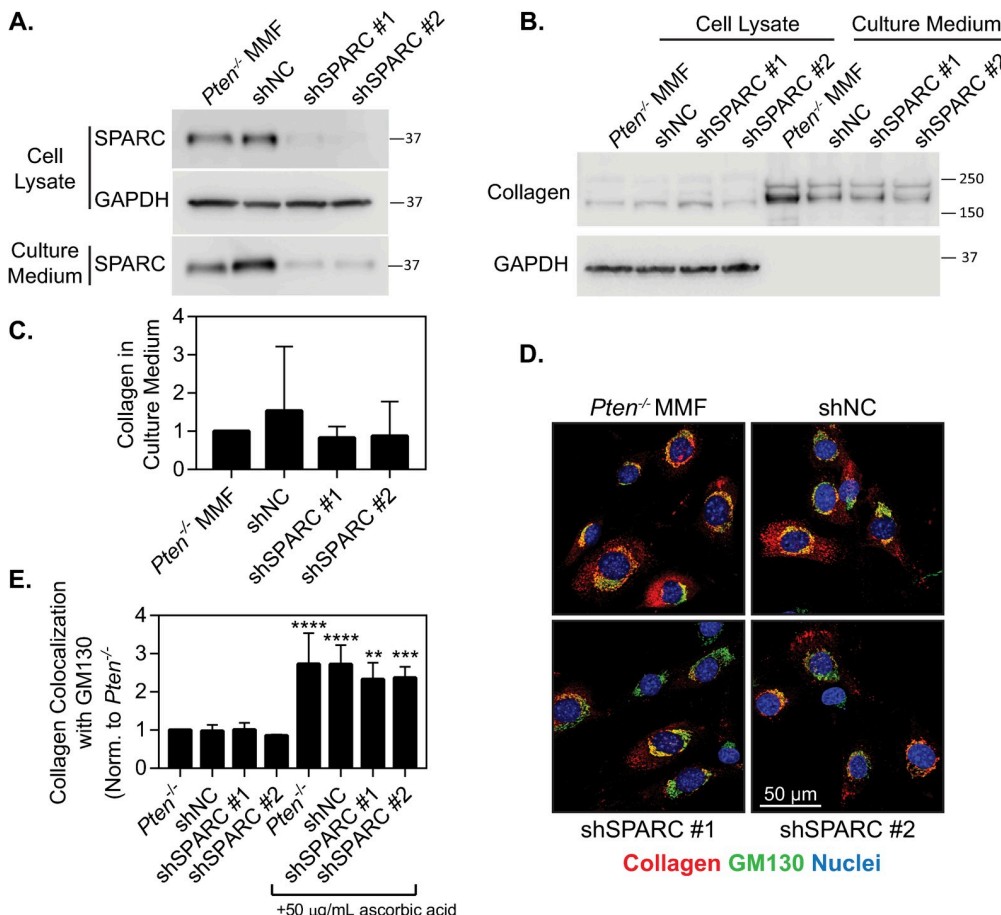

**Fig 5. SPARC knockdown does not affect collagen shuttling.** (A) Western blot verifying SPARC knockdown in both the cell lysate and the culture medium, with GAPDH as a loading control. (B) Western blot for collagen I in the cell lysate and culture medium with SPARC knockdown. (C) Quantification of collagen expression in the culture medium with SPARC knockdown, normalized to *Pten*⁻/⁻ MMF. n = 3+SD, with no significant difference observed between conditions. (D) Immunofluorescence images of MMF plated for 24 hours and cultured with 50 μg/mL ascorbic acid for 1 hour, then fixed and stained for collagen, GM130, and nuclei. (E) Quantification of collagen colocalization with GM130, determined by Manders' correlation coefficient and normalized to the *Pten*⁻/⁻ parental condition. n = 3+SD. **p<0.01, ***p<0.001, ****p<0.0001 compared to the same cell line without ascorbic acid treatment.

cellularity in each image (Fig 6B). Cells modified with shSPARC#1 significantly decreased fiber formation relative to the parental *Pten*⁻/⁻ line. The second SPARC shRNA resulted in more variable fiber formation, but the average was similar to that of the first shRNA sequence. This suggests that SPARC contributes to the increased ability of *Pten*⁻/⁻ cells to form collagen fibers.

To determine whether the effects of SPARC on matrix deposition are specific to collagen or also affect other matrix proteins, murine mammary fibroblasts were incubated with fluorescently labeled fibronectin in a similar manner to the collagen studies. After 24 hours, there was a notable decrease in fibronectin assembly by SPARC knockdown there were notable increases in fiber formation by *Pten*⁻/⁻ fibroblasts as compared to WT fibroblasts, which were abrogated (Fig 6A). Quantification of the fiber area relative to nuclear area in each image revealed a similar trend to collagen (Fig 6C), indicating that SPARC likely plays a role in overall matrix assembly. One possible mechanism SPARC may influence matrix assembly is through fibrillar adhesions, which are critical for the assembly of FN and collagen fibrils [48, 49].

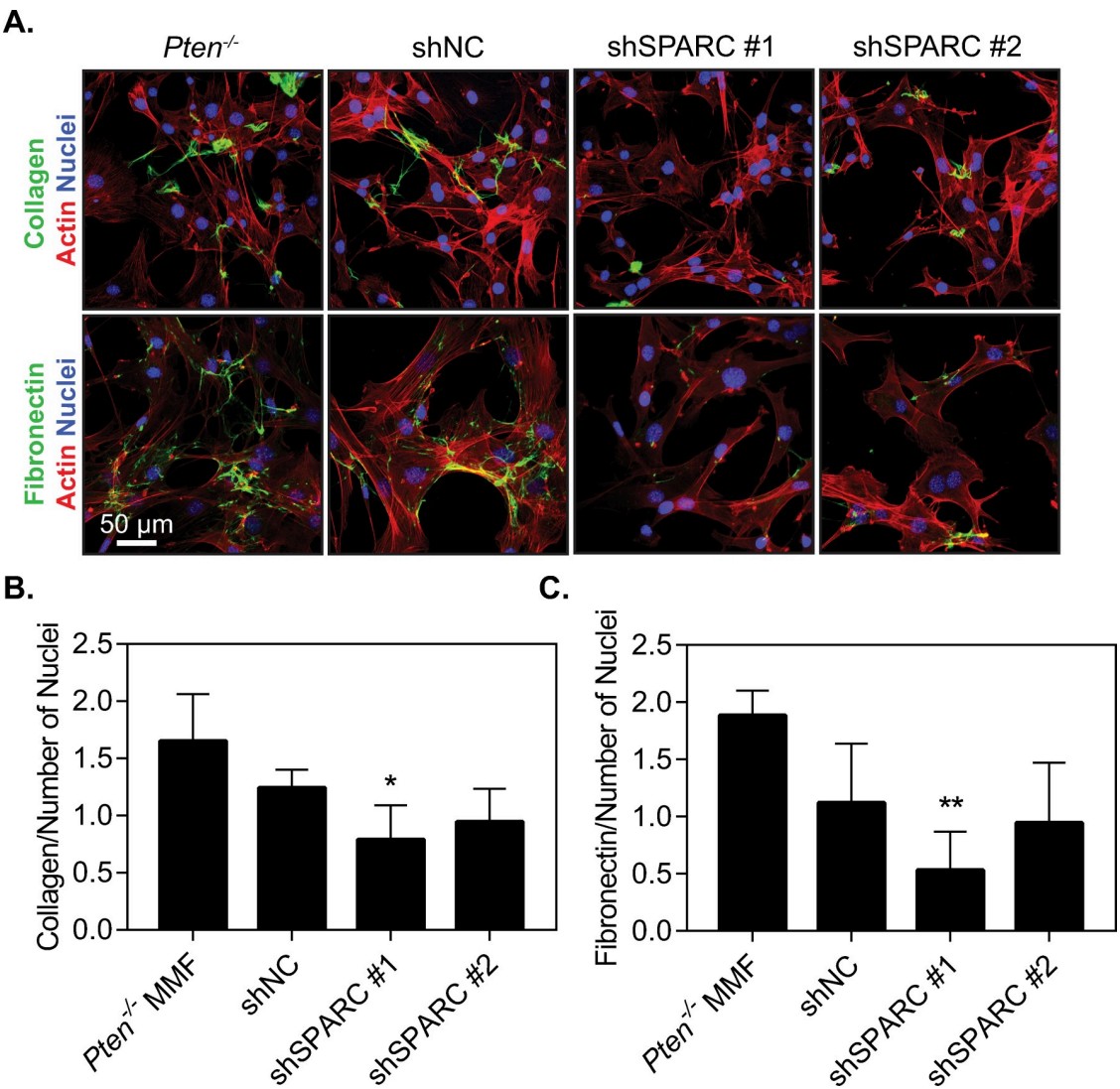

**Fig 6. SPARC expression induced by *Pten* knockout promotes collagen and fibronectin fiber formation.** (A) Immunofluorescence images of cells incubated with fluorescently labeled collagen and fibronectin for 24 hours, then fixed and stained for actin and nuclei. (B) Quantification of the area of each image covered by collagen fibers normalized to number of nuclei in each image. (C) Quantification of the area of each image covered by fibronectin fibers normalized to number of nuclei in each image. n = 3+SD. *p<0.05 relative to *Pten*-/-. **p<0.01 relative to *Pten*-/-.

Immunofluorescence staining of the fibrillar adhesion marker tensin I indicated an increase in the number of fibrillar adhesions in *Pten*-/- as compared to the WT fibroblasts, and knockdown of SPARC decreased the number of fibrillar adhesions (Fig 7).

Matrix assembly is regulated in part by cell contractility, and SPARC has previously been observed to regulate cell contractility pathways through ILK [50]. Furthermore, we have previously shown that *Pten* knockout in fibroblasts increases extracellular matrix alignment through increases in cell contractility [10]. We therefore hypothesized that changes in traction forces might be mediating the effects of SPARC on matrix alignment. The traction forces generated by these cells were examined using traction force microscopy (S6 Fig). Surprisingly, although *Pten* knockout significantly increased the average traction force as well as the net contractile moment generated by cells, SPARC knockdown did not significantly affect either of these

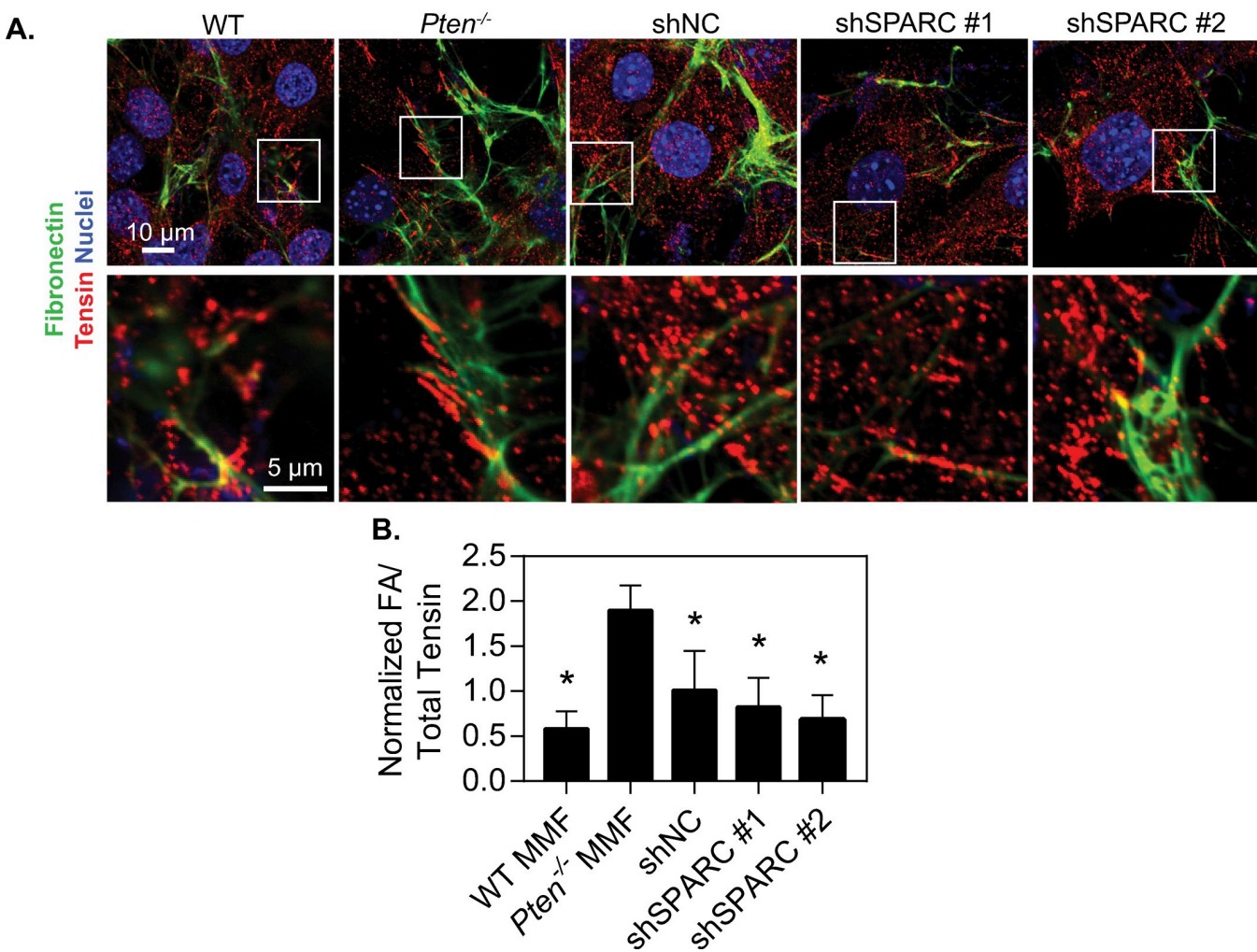

**Fig 7. SPARC expression induced by *Pten* knockout promotes fibrillar adhesion formation.** (A) Top row: Immunofluorescence images of cells incubated with fluorescently labeled fibronectin for 24 hours, then fixed and stained for tensin and nuclei. Inset boxes indicate areas of interest, magnified in bottom row. (B) Quantification of the number of fibrillar adhesions (FAs) per field of view (FOV) relative to the total amount of tensin signal present. n = 3+SD. #p<0.05 relative to *Pten*⁻/⁻.

metrics. This suggests that SPARC plays a role in matrix assembly and matrix alignment independent of traction-force mediated matrix assembly.

## SPARC knockdown mitigates *Pten* knockout-induced matrix alignment

Extracellular matrix alignment surrounding tumors is thought to provide highways for cancer cell invasion away from the primary tumor and is an independent negative prognostic factor in breast cancer patients [51, 52]. *Pten* knockout in mammary fibroblasts has previously been shown to increase matrix alignment both *in vivo* and *in vitro* [10]. As SPARC knockdown abrogated increases in both collagen and fibronectin fiber formation caused by *Pten* knockout, we hypothesized that SPARC would additionally play a role in matrix organization. To test this hypothesis, fibroblast-derived extracellular matrices were generated and imaged using confocal microscopy (Fig 8A). The orientation of the fibers was quantified using OrientationJ software [25] and alignment was determined by taking the fraction of fibers falling within 10˚ or 20˚ of the mode orientation for each image. Using this metric, a higher fraction of fibers falling

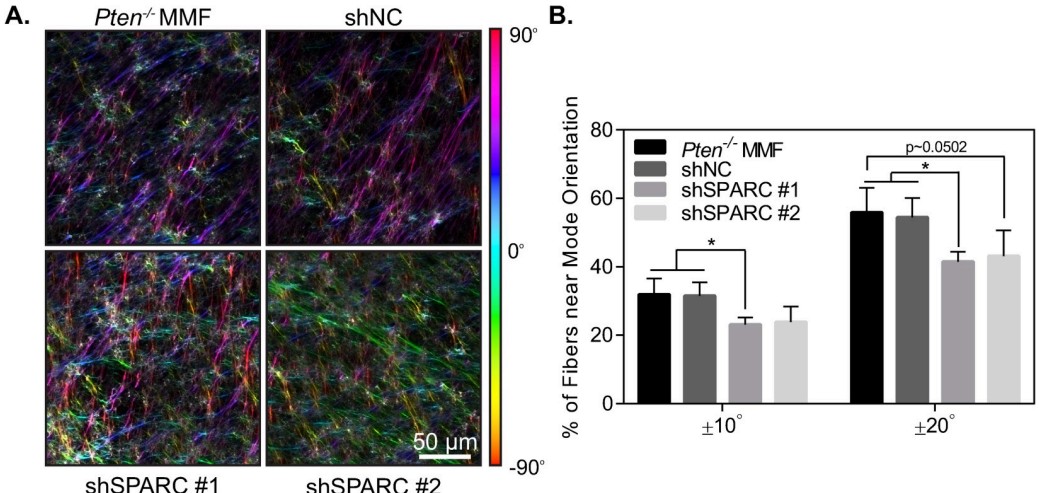

**Fig 8. SPARC knockdown decreases extracellular matrix alignment.** (A) Fibroblast-derived extracellular matrices, color-coded by fiber orientation. (B) Quantification of fibroblast-derived matrix alignment, n = 3+SD.

within this range signifies a more highly aligned matrix. There was a significant decrease in matrix alignment with the first SPARC shRNA, while the second shRNA sequence showed a similar trend but did not reach statistical significance (Fig 8B). Overall, this suggests that SPARC contributes to alignment of the ECM by *Pten*⁻/⁻ cells.

## SPARC expression is upregulated in the tumor microenvironment

While low stromal PTEN expression has previously been observed in breast cancer patients [9], changes in SPARC expression in the stroma of human breast tumors had not been investigated. Using several patient datasets that specifically examined gene expression in the breast stroma (Finak *et al* [27], Karnoub *et al* [28], and Ma *et al* [29]), the expression of *SPARC* in normal and tumor breast stroma was queried. *SPARC* was consistently upregulated in the tumor stroma as compared to the normal stroma in each of these datasets (p 9A, 9B and 9C), underscoring its potential to contribute to collagen deposition in the tumor microenvironment.

## Discussion

Collagen deposition is one of the defining characteristics of high mammographic density and plays a significant role in both tumor formation and cancer progression. We have previously

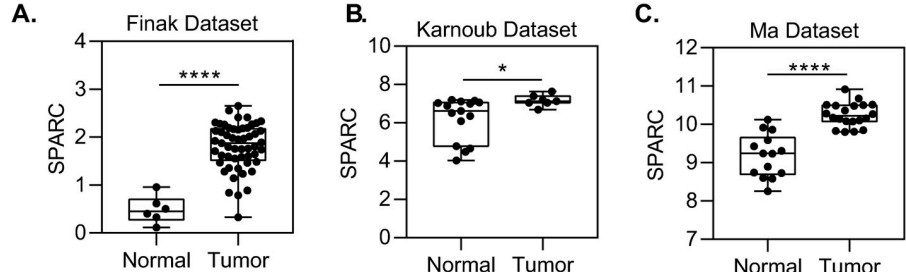

**Fig 9. SPARC expression is upregulated in the tumor microenvironment.** Stromal gene expression of SPARC in normal or breast cancer patients from (A) Finak et al, (B) Karnoub et al, or (C) Ma et al. *p<0.05, ***p<0.001, ****p<0.0001.

shown that low stromal PTEN expression is associated with increases in mammographic density [10] and stromal *Pten* knockout increased collagen deposition in the mammary gland in a mouse model [9]. However, it remained unclear how PTEN loss altered collagen synthesis pathways. Here, we show that *Pten* loss contributes to collagen shuttling out of the cell as well as collagen fiber formation, and that the increases in collagen fiber formation are dependent on increased expression of SPARC. Although SPARC has previously been suggested to act as a collagen chaperone [11], in this *Pten* knockout model SPARC did not affect collagen shuttling. The effects of SPARC on collagen deposition may primarily impact fibrillogenesis, as seen here, or other changes in collagen shuttling due to *Pten* knockout in this model may be masking the effects of SPARC. Collagen shuttling may be dependent on hydroxylation by P4HA3, as we observed significant upregulation of mRNA expression of this enzyme in the PTEN null cells. Another potential mechanism by which SPARC may regulate collagen deposition is by modulating the activity of transglutaminases, a family of proteins involved in extracellular collagen fiber assembly [53]. In dental tissue, transglutaminase crosslinking of collagen was found to be greater in SPARC-null tissues in comparison to WT, suggesting that SPARC bound to collagen may reduce accessibility of collagen to TG modification [54]. TG expression has also been shown to play an important role in the motile and invasive phenotype of metastatic breast cancer [55] and to regulate interactions between cancer cells and the extracellular matrix [56, 57]. Higher levels of tissue TG have been also observed in metastatic lymph node tumors as compared to the primary breast tumors [58], further supporting the role of TG in cancer metastasis.

SPARC expression is widely associated with breast cancer, as approximately 75% of invasive breast cancers show strong SPARC expression, while in normal mammary tissue 92% of samples show undetectable or poorly detectable expression [59]. The role of SPARC in breast cancer remains in question, as it has been linked both to increased progression [60] and decreased cancer cell proliferation [61, 62]. In triple negative breast cancer patients, high SPARC expression correlates with poor overall survival [63]. These differential effects may also be linked to diverging roles of SPARC in the tumor as compared to the stroma, as stromal expression but not tumor expression has been linked to poor patient outcomes in pancreatic ductal adenocarcinoma [64]. Here, we show that stromal expression of SPARC is significantly upregulated in breast cancer as compared to the normal breast in several different patient cohorts. Further studies to compare the effects of SPARC in the tumor and the stroma are warranted.

Consistent with the idea that stromal SPARC expression may promote cancer progression, SPARC knockdown in fibroblasts decreased ECM fiber formation and alignment in our study, and these factors play a large role in tumor progression. Periductal collagen alignment is associated with high mammographic density [10, 65] and collagen alignment perpendicular to the tumor edge can form "highways" for cancer cell invasion [51]. The formation of perpendicularly-oriented collagen fibers is an independent prognostic factor in breast cancer patients [52] and we have previously observed that *Pten* knockout in fibroblasts increases the formation of these fibers surrounding *MMTV-Neu*-generated tumors in mice [10]. Therefore, SPARC expression in fibroblasts may support cancer progression, despite its apparent antiproliferative effects in cancer cells. We suggest that stromal SPARC plays a role in regulating ECM production and alignment in breast cancer, and that further work should be done to characterize the role that SPARC plays in the stroma as compared to the tumor.

## Supporting information

**S1 Fig. Uncropped Western blot images for blots used in Figs 2, 3 and 4.**
(PDF)

**S2 Fig. Fibrillar adhesion length analysis.** Dotted line indicates threshold used for minimum length of fibrillar adhesions.
(TIF)

**S3 Fig. Collagen immunofluorescence and phalloidin staining of cells cultured with ascorbic acid.**
(TIF)

**S4 Fig. Quantification of SPARC knockdown.** (A) Quantification of SPARC expression in the cell, normalized to GAPDH as a loading control. (B) Quantification of SPARC in the culture medium. n = 3+SD, $^{*}$p$<$0.05, $^{**}$p$<$0.01.
(TIF)

**S5 Fig. Isolated images of fluorescently labeled matrix fibers.** (A) Collagen fibers assembled by fibroblasts. (B) Fibronectin fibers assembled by fibroblasts.
(TIF)

**S6 Fig. SPARC knockdown does not affect cell traction forces.** (A) Average traction force per cell, shown as mean±SD with values of individual cells plotted. (B) Net contractile moment per cell. n = 25–46 cells. $^{*}$p$<$0.05, $^{***}$p$<$0.001, $^{****}$p$<$0.0001.
(TIF)

## Acknowledgments

Images presented in this report were generated using the instruments and services at the Campus Microscopy and Imaging Facility, The Ohio State University.

## Author Contributions

**Conceptualization:** Caitlin E. Jones, Joe T. Sharick, Sheila E. Colbert, Steven T. Sizemore, Jennifer L. Leight.

**Data curation:** Caitlin E. Jones, Joe T. Sharick, Vasudha C. Shukla, Samir N. Ghadiali, Steven T. Sizemore.

**Formal analysis:** Caitlin E. Jones, Joe T. Sharick, Sheila E. Colbert, Vasudha C. Shukla, Joshua M. Zent, Samir N. Ghadiali, Steven T. Sizemore, Jennifer L. Leight.

**Investigation:** Caitlin E. Jones, Joe T. Sharick, Sheila E. Colbert, Vasudha C. Shukla, Michael C. Ostrowski, Samir N. Ghadiali, Steven T. Sizemore, Jennifer L. Leight.

**Methodology:** Caitlin E. Jones, Joe T. Sharick, Joshua M. Zent, Michael C. Ostrowski, Jennifer L. Leight.

**Project administration:** Jennifer L. Leight.

**Resources:** Jennifer L. Leight.

**Supervision:** Jennifer L. Leight.

**Validation:** Caitlin E. Jones.

**Visualization:** Caitlin E. Jones.

**Writing – original draft:** Caitlin E. Jones, Joe T. Sharick, Vasudha C. Shukla, Jennifer L. Leight.

**Writing – review & editing:** Caitlin E. Jones, Joe T. Sharick, Vasudha C. Shukla, Joshua M. Zent, Michael C. Ostrowski, Samir N. Ghadiali, Steven T. Sizemore, Jennifer L. Leight.

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
