## [Decision Letter · Decision Letter 0]

10 Jun 2020

PONE-D-20-14527

Pten regulates collagen fibrillogenesis by fibroblasts through SPARC

PLOS ONE

Dear Dr. Leight,

Thank you for submitting your manuscript to PLOS ONE. After careful consideration, we feel that it has merit but does not fully meet PLOS ONE’s publication criteria as it currently stands. Therefore, we invite you to submit a revised version of the manuscript that addresses the points raised during the review process.

We look forward to receiving your revised manuscript.

Kind regards,

Donald Gullberg, PhD

Academic Editor

PLOS ONE

Journal Requirements:

2. At this time, we request that you  please report additional details in your Methods section regarding animal care, as per our editorial guidelines:

(1) Please state the source and number of mice used in the study  

(2) Please provide details of animal welfare (e.g., shelter, food, water, environmental enrichment)

(3) Please include the method of euthanasia  

Thank you for your attention to these requests.

3. Please provide additional information about each of the cell lines used in this work, including source any quality control testing procedures (authentication, characterisation, and mycoplasma testing). For more information, please see http://journals.plos.org/plosone/s/submission-guidelines#loc-cell-lines.

4. In the Methods section, please provide the accession numbers of the publicly available patient datasets analyzed in the study.

'Images presented in this report were generated using the instruments and services at the Campus

Microscopy and Imaging Facility, The Ohio State University. This facility is supported in part by NIH

cancer center grant P30 CA016058 to The Ohio State University, National Cancer Institute, Bethesda,

MD.'

'The author(s) received no specific funding for this work.'

Additional Editor Comments (if provided):

Two experts in the field have taken part of your work. Based on their comments the editorial decision is " major revision". Please study their comments carefully to address their points. I look forward to a revised improved version.

Reviewers' comments:

Reviewer's Responses to Questions

**Comments to the Author**

1. Is the manuscript technically sound, and do the data support the conclusions?

Reviewer #1: Partly

Reviewer #2: Yes

2. Has the statistical analysis been performed appropriately and rigorously? 

Reviewer #1: Yes

Reviewer #2: Yes

3. Have the authors made all data underlying the findings in their manuscript fully available?

Reviewer #1: Yes

Reviewer #2: Yes

4. Is the manuscript presented in an intelligible fashion and written in standard English?

Reviewer #1: Yes

Reviewer #2: Yes

5. Review Comments to the Author

Reviewer #1: The manuscript investigates on how PTEN loss contributes to increased collagen deposition. The authors have used murine mammary fibroblasts from Wildtype (PtenloxP/loxP) and Pten-/- (Fsp-Cre;Ptenloxp/loxp) mice. The authors suggest that Pten loss contributes to increased collagen deposition by increased shuttling of collagen out of the cell. Further, the authors also suggest that Pten loss mediated upregulation of SPARC (Secreted protein, acidic and rich in cysteine) regulates collagen fibre formation and assembly. In general, the manuscript is technically sound with some novel findings and some of the claims are supported with experimental data. However, few points remain to be addressed.

Major comments:

1. In Figure 2B and 4B, it is unclear why there is no band for collagen in Pten KO cell lysates, despite collagen staining in 2C and 4D. Could this be due to insufficient solubilization by RIPA lysis buffer ? Could the authors use more stringent lysis buffer to measure collagen that is deposited on to the matrix

2. In Figure 4B, the migration of bands for collagen are different from figure 2B. Particularly, the top band around 200 kDa, which seems to be increased in shSPARC 1 and shSPARC 2. Do these bands indicate defect in collagen deposition to the matrix in the absence of SPARC and therefore increased in the medium ? Could the authors use more stringent lysis buffer to measure collagen that is deposited on to the matrix.

3. In figure 5A, SPARC knockdown cells seems to have more exogenous fibronectin fibrils than WT cells. Could the authors stain for endogenous cellular fibronectin or fibronectin-fibrillar adhesion markers such as integrin α5 or integrin β1 or Tensin to see if cell-fibronectin interactions are altered in SPARC knockdown cells.

Minor comments:

1. In Figure 2C, does the authors observe any significant difference in cell spreading or cell adhesion dynamics and signaling between WT and Pten KO cells ?

2. In figure 3C & 4C, what was the loading control used in the western blot quantification for culture medium samples ?

Reviewer #2: The manuscript by Jones et al. describes the characterization of abrogated expression of PTEN in stromal fibroblasts in a murine model of breast cancer. The authors interrogate differences in collagen production, secretion, and assembly influenced by decreases in PTEN, as decreases in PTEN are associated with poor prognosis and increases in collagen density. The authors present evidence for increases in the matricellular protein SPARC, driven by decreases in PTEN, providing a mechanistic basis for increases in collagen assembly. Whereas PTEN-dependent differences in collagen shuttling within the cell were not significantly altered by altered expression of SPARC, significant changes in collagen and fibronectin assembly were observed in PTEN -/- cells with decreased SPARC production. The manuscript is interesting and contributes significantly to the field. Issues to be addressed prior to publication are listed below:

1). The authors should state the number of primary cell isolates that contribute to each figure. For example, does the n number provided reflect separate isolates of primary cells or separate experiments with one primary isolation. The number of mice and/or mammary glands contributing to each primary isolate should be stated.

2). It is curious that collagen staining by immunohistochemistry is evident in cell layers as shown in Fig 2C and Fig.4D in PTEN -/- cells but no detection is seen in cell lysates by western blot analysis. Do the authors have an explanation for this? Also, the MW for procollagen in Fig 2B is not consistent with that in Fig.4B. Are the origin of these differences known? Was ascorbate used in the experiments from which the collagen was measured in layers and in media?

3). Overall, the authors should consider the contribution of SPARC-dependent changes in transglutaminase activity on collagen assembly in this milieu. As the collagen fiber diameter shown in Fig. 1D suggests that decreases in PTEN (and hence increases in SPARC) are associated with thicker collagen fibers. Trombetta-eSilva et al. (JBMR, 2015) and Moore Rosset et al. (JDR 2020) demonstrated that smaller collagen fibers indicative of SPARC-null tissues are reversed with inhibition of transglutaminase activity. Hence, increases in collagen fibers seen in PTEN -/- might result from increases in SPARC and modulation of transglutaminase activity. Did the authors measure levels of transglutaminase expression by mRNA? At the least, this might be added to the discussion.

6. PLOS authors have the option to publish the peer review history of their article (what does this mean?). If published, this will include your full peer review and any attached files.

Reviewer #1: No

Reviewer #2: No

---

## [Author Response · Author response to Decision Letter 0]

16 Dec 2020

We thank the reviewers for their thoughtful comments and insights. We have revised the manuscript to address their comments and provide detailed responses below. 

Reviewer #1:

Major comments:

1. In Figure 2B and 4B, it is unclear why there is no band for collagen in Pten KO cell lysates, despite collagen staining in 2C and 4D. Could this be due to insufficient solubilization by RIPA lysis buffer ? Could the authors use more stringent lysis buffer to measure collagen that is deposited on to the matrix.

Author response: The differences observed in the amount of collagen may be due to the difference in timing between the immunofluorescence (1 hr of ascorbic acid treatment) and western blot (20 hr of ascorbic acid treatment) experiments. We have made the treatment times more clear in the methods and results section, as well as included new data with the immunofluorescence staining for collagen at the 0, 1, and 20 hr ascorbic acid treatment (Supplemental Figure 5), which indicates decreased collagen inside of the cells at 20 hr. This immunofluorescence staining also indicates a lack of collagen matrix deposition outside the cells at the 20 hr time point, as indicated by lack of pericellular staining. Therefore, we conclude it is unlikely that we are not solubilizing deposited matrix since little to none has been deposited at this time point. Furthermore, our lysis conditions are similar to established protocols that have been used to solubilize deposited matrix at similar time points (~24 hrs) (Soucy and Romer, Matrix Biology 2009).

2. In Figure 4B, the migration of bands for collagen are different from figure 2B. Particularly, the top band around 200 kDa, which seems to be increased in shSPARC 1 and shSPARC 2. Do these bands indicate defect in collagen deposition to the matrix in the absence of SPARC and therefore increased in the medium ? Could the authors use more stringent lysis buffer to measure collagen that is deposited on to the matrix.

Author response: The bands likely appear different due to differences in electrophoresis running times, where sometimes two distinct bands are apparent, where as with shorter run times it appears as if there is only one band. We have re-ran some of the old lysates and prepared new lysates and have been more consistent with electrophoresis times. Now, we consistently observe two bands for the upper MW bands in the collagen blots (Fig 2 and Fig 4, Supplemental Figures). 

3. In figure 5A, SPARC knockdown cells seems to have more exogenous fibronectin fibrils than WT cells. Could the authors stain for endogenous cellular fibronectin or fibronectin-fibrillar adhesion markers such asintegrin α5 or integrin β1 or Tensin to see if cell-fibronectin interactions are altered in SPARC knockdown cells.

Author response: We have included an additional supplemental figure (S6), in which we have included only the fibronectin or collagen channel so that one can see this signal separate from the actin staining. We have also immunostained for the fibrillar adhesion marker tensin to investigate cell-fibronectin interactions, included in a new figure (Figure 7). We observe more fibrillar adhesions in the Pten KO fibroblasts as compared to wildtype cells, and a decrease in the adhesion number with SPARC knockdown. 

Minor comments:

1. In Figure 2C, does the authors observe any significant difference in cell spreading or cell adhesion dynamics and signaling between WT and Pten KO cells ?

Author response: Yes, the PTEN KO fibroblasts spread more and the nuclei appear larger than the WT cells at the same magnification, which also agrees with a previous publication (Jones et al., Neoplasia 2019). 

2. In figure 3C & 4C, what was the loading control used in the western blot quantification for culture medium samples ?

Author response: Equal amounts of concentrated medium were loaded for the western blots, and we verify similar loading and protein transfer with Ponceau S staining. 

Reviewer #2.

1). The authors should state the number of primary cell isolates that contribute to each figure. For example, does the n number provided reflect separate isolates of primary cells or separate experiments with one primary isolation. The number of mice and/or mammary glands contributing to each primary isolate should be stated.

Author response: The fibroblast cells were not primary cell isolates but a spontaneously immortalized cell line previously generated in Trimboli et al., Nature 2009. We have clarified the methods to indicate that these were not primary cell isolates. N in cell culture experiments indicates the number of separate experiments that were performed, but all were with the same original cell line.

2). It is curious that collagen staining by immunohistochemistry is evident in cell layers as shown in Fig 2C and Fig.4D in PTEN -/-cells but no detection is seen in cell lysates by western blot analysis. Do the authors have an explanation for this? Was ascorbate used in the experiments from which the collagen was measured in layers and in media?

Author response: The lack of collagen bands in the original western images were likely due to the timing of the ascorbic acid treatment. Please see response to Reviewer 1, Major Comment 1. To observe collagen transport from the ER to the Golgi within the cells, cells were treated with ascorbic acid for one hour prior to fixation. For the western blots, cells were treated with ascorbic acid for 20 hr. The collagen immunostaining in supplemental figure 5 also shows lower collagen at the 20 hr treatment as compared to the 1 hr. The ascorbic acid treatment of cells lysed for western blots was mistakenly left out of the original submission, we have now corrected this in the methods and the results section. 

Also, the MW for procollagen in Fig 2B is not consistent with that in Fig.4B. Are the origin of these differences known? 

Author response: Please see Reviewer #1, Comment #2. 

3). Overall, the authors should consider the contribution of SPARC-dependent changes in transglutaminase activity on collagen assembly in this milieu. As the collagen fiber diameter shown in Fig. 1D suggests that decreases in PTEN (and hence increases in SPARC) are associated with thicker collagen fibers. Trombetta-eSilva et al. (JBMR, 2015) and Moore Rosset et al. (JDR 2020) demonstrated that smaller collagen fibers indicative of SPARC-null tissues are reversed with inhibition of transglutaminase activity. Hence, increases in collagen fibers seen in PTEN -/-might result from increases in SPARC and modulation of transglutaminase activity. Did the authors measure levels of transglutaminase expression by mRNA? At the least, this might be added to the discussion.

Author response: We thank the reviewer for identifying this interesting potential mechanism. Because there are multiple transglutaminase family members that may contribute differently to our observations, examining transglutaminase is beyond the scope of this manuscript, however we have added this to the discussion.

---

## [Decision Letter · Decision Letter 1]

6 Jan 2021

Pten regulates collagen fibrillogenesis by fibroblasts through SPARC

PONE-D-20-14527R1

Dear Dr. Leight,

We’re pleased to inform you that your manuscript has been judged scientifically suitable for publication and will be formally accepted for publication once it meets all outstanding technical requirements.

Kind regards,

Donald Gullberg, PhD

Academic Editor

PLOS ONE

Additional Editor Comments (optional):

Both reviewers are satisfied with your revised version. Please see two additional minor comments from reviewers ( typo + labelling of collagen chains) that I trust authors will fix.

Reviewers' comments:

Reviewer's Responses to Questions

**Comments to the Author**

1. If the authors have adequately addressed your comments raised in a previous round of review and you feel that this manuscript is now acceptable for publication, you may indicate that here to bypass the “Comments to the Author” section, enter your conflict of interest statement in the “Confidential to Editor” section, and submit your "Accept" recommendation.

Reviewer #1: All comments have been addressed

Reviewer #2: (No Response)

2. Is the manuscript technically sound, and do the data support the conclusions?

Reviewer #1: Yes

Reviewer #2: Yes

3. Has the statistical analysis been performed appropriately and rigorously? 

Reviewer #1: (No Response)

Reviewer #2: Yes

4. Have the authors made all data underlying the findings in their manuscript fully available?

Reviewer #1: Yes

Reviewer #2: Yes

5. Is the manuscript presented in an intelligible fashion and written in standard English?

Reviewer #1: Yes

Reviewer #2: Yes

6. Review Comments to the Author

Reviewer #1: In the current manuscript, the authors have made great improvement with satisfactory data.

Typo:

Line 115 and 429: Tensin 1

Reviewer #2: The Authors have done a nice job of responding to the previous concerns. This reviewer would only ask one point to address prior to publication: In FIg. 2B, the Arrows indicating pro collagen and the alpha chains of collagen do not designate bands, particularly in the case of the alpha subunits, the arrow lines up with no bands. In our experience, alpha subunits generally run between 150 and 100 MW bands. The bands in this blot and in Figure 5B appear to run higher and therefore are likely to be pro collagen and procollagen intermediates although it is appreciated that collagen, as a non-globular protein can run differently in distinct gel systems. My recommendation would be to not attempt to designate specific bands (as done in Fig. 5B) and perhaps note in the legend that bands represent pro collagen, processing intermediates, and alpha subunits.

7. PLOS authors have the option to publish the peer review history of their article (what does this mean?). If published, this will include your full peer review and any attached files.

Reviewer #1: No

Reviewer #2: **Yes: **Amy Bradshaw

---

## [Editor Report · Acceptance letter]

12 Jan 2021

PONE-D-20-14527R1 

*Pten* regulates collagen fibrillogenesis by fibroblasts through SPARC 

Dear Dr. Leight:

I'm pleased to inform you that your manuscript has been deemed suitable for publication in PLOS ONE. Congratulations! Your manuscript is now with our production department. 

Kind regards, 

on behalf of

Professor Donald Gullberg 

Academic Editor

PLOS ONE